# An Efficient and Secure Blockchain Consensus Protocol for Internet of Vehicles

Xueming Si [1,2,3], Min Li [1,2,3,*], Zhongyuan Yao [1,*], Weihua Zhu [1,2,3], Jianmei Liu [1] and Qian Zhang [4]

1    Frontier Information Technology Research Institute, Zhongyuan University of Technology, Zhengzhou 450007, China; 9770@zut.edu.cn (X.S.); 9773@zut.edu.cn (W.Z.); 6933@zut.edu.cn (J.L.)
2    Henan International Joint Laboratory of Blockchain and Data Sharing, Zhengzhou 450007, China
3    Henan Key Laboratory of Network Cryptography Technology, Zhengzhou 450007, China
4    College of Computer Science, Zhongyuan University of Technology, Zhengzhou 450007, China; 6382@zut.edu.cn
*    Correspondence: 2021116578@zut.edu.cn (M.L.); yaozhongyuan@zut.edu.cn (Z.Y.)

**Abstract:** Conventional blockchain consensus protocols tailored for the Internet of Vehicles (IoV) usually face low transaction throughput, high latency, and elevated communication overhead issues. To address these issues, in this paper, we propose ESBCP, an efficient and secure blockchain consensus protocol for the IoV environment. Firstly, considering the significant performance differences among nodes in the IoV, we designed a blockchain consensus model for the IoV. Roadside units execute a trust evaluation mechanism to select high-quality vehicle nodes for the consensus process, thereby reducing the likelihood of malicious nodes in the consensus cluster. Secondly, we designed a node partition strategy to adapt to the dynamic feature of the IoV. Finally, addressing the mobility of nodes in the IoV, we introduced a dynamic unique node list. Vehicle nodes can promptly select nodes with high reliability from the list of communicable nodes to join their unique node list, while also promptly removing nodes with low reliability from their unique node list. Combining these strategies, we propose DK-PBFT, an improved Practical Byzantine Fault Tolerance consensus algorithm. The algorithm meets the efficiency and mobility requirements of vehicular networks. Through theoretical analysis, ESBCP could prevent external and internal security risks while reducing communication overhead. Experimental verification demonstrated that ESBCP effectively reduces consensus latency and improves transaction throughput. Our proposed ESBCP can be used in other application scenarios that require high consensus efficiency.

**Keywords:** blockchain; consensus protocols; IoV; trust evaluation mechanism; node partition strategy; dynamic unique node list

## 1. Introduction

### 1.1. Research Backgrounds

The Internet of Vehicles (IoV) is an open and integrated network that collaborates among vehicles, roads, people, and roadside infrastructure. It facilitates timely information sharing and cooperation between vehicles by integrating various information technologies [1]. The IoV plays a crucial role in the real world by enabling real-time communication and information sharing. It significantly contributes to areas such as traffic safety, traffic flow optimization, navigation, and emergency assistance. It brings a multitude of conveniences and improvements to the real-world transportation system. In 2020, German artist S. Weckert manipulated the navigation system of Google by slowly walking through the streets with a cart containing 99 mobile phones, tricking the entire Google navigation network into believing that the area was congested with traffic. This incident highlighted the potential security risks associated with such tactics, as research suggests that hackers could exploit this method to induce changes in traffic flow. Currently, the IoV still faces challenges such as inadequate data security assurance, potential leakage of user privacy,

and vulnerability of nodes. These issues indirectly hinder the widespread application and technological development of the IoV. Blockchain has emerged as an effective solution for addressing several such issues. With the technological features of decentralization, anonymity, tamper resistance, and traceability, blockchain provides a solution for solving or alleviating security and device management issues in the IoV. However, the IoV exhibits characteristics such as a large-scale node population, dynamic changes in node locations, and mostly resource-constrained environments. Directly incorporating blockchain technology into the IoV would inevitably impact the efficiency of blockchain consensus, leading a decrease in overall system performance. This would fail to meet the practical requirements of the IoV for high real-time performance and reliability. Therefore, proposing an efficient and secure blockchain consensus protocol tailored for the IoV environment becomes an urgent and critical issue that needs attention and resolution.

### 1.2. Related Work

Currently, there is relatively limited research on optimizing blockchain consensus protocols for the IoV. Considering the vulnerability of nodes in the IoV and the presence of malicious behavior, Byzantine Fault Tolerance (BFT) consensus algorithms, which possess Byzantine fault tolerance capabilities, are more suitable for IoV scenarios compared to other algorithms. Although the classical Practical Byzantine Fault Tolerance (PBFT) algorithm allows for a proportion of Byzantine nodes to be less than 1/3 of all nodes, its efficiency sharply declines as the number of network nodes increases. Therefore, a substantial amount of improvement work has been proposed for PBFT. Generally, optimization strategies for PBFT algorithms can be roughly categorized into three types. The first type is controlling the number of participating consensus nodes. This category selects a subset of all system nodes as a committee. The committee nodes participate in the consensus process and then sends the consensus result to the remaining nodes, reducing communication overhead to enhance system scalability. The second type is optimizing the consensus process. This involves improving the selection method of the primary node or adding node evaluation mechanisms to reduce unnecessary communication, thereby increasing consensus efficiency. The third type is enhancing the consensus structure. This category introduces layered models or grouping concepts to reduce the number of required consensus communications, thereby improving consensus efficiency.

In terms of controlling the number of consensus nodes, ref. [2] proposes the VPBFT (PBFT based on Voting) consensus algorithm. It primarily focuses on categorizing nodes in the network into four types using a voting mechanism and assigning them different permissions to reduce the number of participating nodes in the consensus process. However, it does not consider the issue of communication overhead between nodes. Ref. [3] introduces the EPBFT (Extensible PBFT) consensus algorithm, which is designed for dynamic networks. This algorithm utilizes Verifiable Random Function (VRF) to select a subset of nodes to participate in the consensus process, thereby reducing the communication volume of PBFT. However, it does not address the communication latency issue caused by node mobility. Ref. [4] presents the G-PBFT (Geographic Practical Byzantine Fault Tolerance) consensus algorithm tailored for Internet of Things (IoT) scenarios. It achieves consensus using the geographical information of fixed devices, selecting nodes with relatively stable positions as endorsers to participate in PBFT consensus in order to reduce the cost of validating and recording transactions. However, it relies heavily on nodes with relatively stable positions, which are more susceptible to attacks and pose potential security risks.

In the aspect of optimizing the consensus process, ref. [5] proposes the OBFT (Optimistic Byzantine Fault Tolerance) consensus algorithm, which ensures the security and liveness of the algorithm by dynamically setting timeout periods, thereby improving its efficiency to a certain extent. However, for large-scale vehicular network scenarios, its scalability is limited. Based on SVM and PBFT, ref. [6] introduces a primary node selection strategy and incorporates a dynamically adjustable trust mechanism to enhance the security of the consensus system. However, it does not significantly improve throughput and

consensus latency. Ref. [7] categorizes transactions into equal and unequal transactions, only publicly disclosing erroneous transactions, thereby reducing the number of consensus iterations and increasing the efficiency and scalability of blockchain consensus. The consensus coordinator used in this approach is centralized, which may face single point of failure issues. Additionally, when the system fails or restarts, this does not address the recovery of the coordinator. Ref. [8] presents the Improved Multi-Primary-Node Practical Byzantine Fault Tolerance (IMPBFT) consensus algorithm, which selects multiple primary nodes to jointly receive client transactions and introduces a pipeline to achieve concurrent execution of IMPBFT consensus, thus improving consensus efficiency. However, it requires an ample amount of resources, such as memory space and computing power for consensus execution, which is not feasible for the limited resources of vehicular network devices. Ref. [9] employs a reputation mechanism calculated through logistic regression to enhance the PBFT consensus process, In addition, it proposes R-PBFT (Reputed PBFT), a fast and intelligent consensus mechanism algorithm. It does not take into account the impact of a large number of mobile nodes in the network on consensus efficiency.

In terms of improving consensus structures, ref. [10] introduces the K-PBFT consensus mechanism, which employs an enhanced k-medoids clustering algorithm to cluster and hierarchically partition consensus nodes, optimizing the consensus process for large-scale consensus nodes. This allows the blockchain model to be applicable in a wider range of scenarios. Ref. [11] conducts node clustering based on location features and decomposes the consensus task to enhance the PBFT consensus algorithm, reducing the required communication for consensus. However, both Refs. [10,11] may have collusion or malicious behavior among nodes within clusters, posing a security risk to the system. Ref. [12] proposes a parallel consensus mechanism based on the DAG lattice structure, addressing the low efficiency caused by an excessive number of consensus nodes and node mobility of PBFT through network sharing. Nevertheless, maintaining the DAG lattice structure incurs high costs and can significantly impact the overall system performance. The DGBFT (Dynamic Grouping Byzantine Fault Tolerance Mechanism) consensus algorithm from ref. [13] groups nodes based on trust levels, effectively excluding malicious nodes and greatly reducing communication complexity. The SG-PBFT consensus algorithm from ref. [14] further improves system consensus efficiency by grouping nodes based on a scoring system. However, the communication complexity in refs. [13,14] remains unchanged compared with PBFT. When the number of system nodes is too high, their substantial communication overhead can greatly hinder consensus efficiency. Ref. [15] improves the PBFT algorithm by electing delegated agents to participate in local and global consensus based on trust levels, thereby enhancing consensus efficiency. Ref. [16] introduces the CDBFT consensus algorithm, dividing nodes into various organizations and selecting a representative node from each organization. This process involves two stages: intra-organizational consensus and representative node consensus, reducing the number of nodes participating in consensus and lowering communication overhead. Ref. [17] proposes C-PBFT (Concurrent PBFT), a two-tier PBFT consensus mechanism. Through an analysis of historical transactions, it divides supply chain nodes into several clusters. Each cluster's primary node is determined through reputation assessment to reduce communication overhead. However, the communication volume in refs. [15–17] remains at a quadratic level. With a large number of nodes, their consensus efficiency sharply decreases, indicating poor scalability.

The above solutions address some of the shortcomings of the classical PBFT consensus algorithm, to some extent improving the efficiency and scalability of the algorithm. However, current optimization work on PBFT algorithms has largely overlooked the specificity of the IoV environment and has not tailored optimizations according to the dynamic and heterogeneous nodes in the IoV. It is evident that existing work can only provide a research direction for how to propose an efficient and secure blockchain consensus mechanism suitable for the IoV environment. Further research is still needed to comprehensively address these issues.

*1.3. Research Contributions*

In this paper, we introduce Efficient Security Blockchain Consensus Protocol (ESBCP) tailored for the IoV. The algorithm addresses the issue of untrustworthy interactions when communicating with different neighboring vehicles through the implementation of a trust assessment mechanism. It incorporates a node partition mechanism to enhance the concurrency of consensus tasks, thereby accelerating the speed of the consensus process. Additionally, the dynamic unique node list ensures adaptability to the dynamic nature of IoV, thereby enhancing consensus stability. In light of this, our contributions in this paper are multifold.

(1) We propose a blockchain consensus model suitable for the IoV environment, employing a permissioned chain mechanism involving entities such as Onboard Units (OBUs) and Roadside Units (RSUs). The consensus process is divided into the Pre-prepare, Prepare1, Prepare2, Commit1, and Commit2 phases.

(2) Based on the above consensus model, we introduce the ESBCP consensus protocol tailored for the IoV environment. This protocol integrates various strategies including trust assessment mechanisms, node partition, Dynamic Unique Node List (DUNL), and improved consensus algorithms. It addresses the high latency and difficult adaptability issues present in the classical PBFT algorithm.

(3) We conduct detailed theoretical analysis and comparative experimental validation of the ESBCP consensus protocol. The theoretical analysis demonstrates the effectiveness of ESBCP in preventing external and internal security risks. The communication complexity of ESBCP is O(n). The protocol exhibits excellent scalability. Comparative experiments indicate that, in contrast to the CDBFT and SG-PBFT algorithms, ESBCP achieves lower latency, higher throughput, and is more suitable for large-scale IoV environments.

## 2. Preliminary Knowledge

*2.1. Internet of Vehicles (IoV)*

The IoV, also known as Vehicle to Everything (V2X), is a significant application of the Internet of Things (IoT) in the field of intelligent transportation. Through onboard devices, it involves real-time data collection from vehicles, including information on their driving status, speed, route, location, and environmental surroundings. This allows for the supervision of all vehicles, ultimately enhancing traffic efficiency. Leveraging modern information and communication technologies such as 5G [18,19], the IoV enables communication between vehicles and various entities, including Vehicle-to-Vehicle (V2V), Vehicle-to-Person (V2P), Vehicle-to-Road (V2R), Vehicle-to-Network (V2N), and Vehicle-to-Infrastructure (V2I), ultimately leading to connectivity between vehicles and everything. Consequently, the IoV not only connects vehicles to one another but also integrates them with pedestrians, roads, networks, and infrastructure, enabling efficient information perception, intelligent analysis, and secure sharing. The architecture of the IoV is illustrated in Figure 1. The IoV possesses characteristics such as real-time interconnectivity, vehicle sensing and recognition, data sharing and exchange, intelligent traffic management, safety and protection, vehicle autonomy, as well as data privacy and security. These features lay the foundation for achieving intelligent transportation, enhancing traffic safety, and providing more convenient travel experiences [20,21].

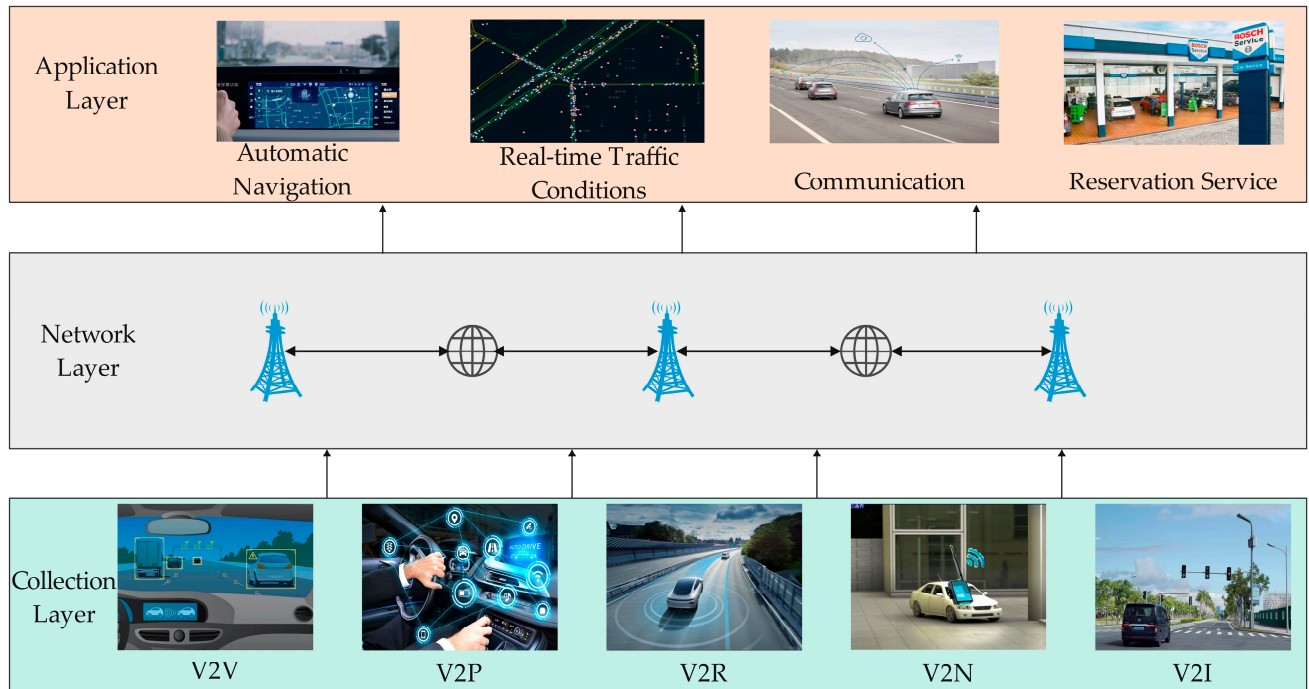

**Figure 1.** Architecture of the IoV.

### 2.2. Blockchain

Blockchain technology was initially introduced in a paper entitled "Bitcoin: A Peer-to-Peer Electronic Cash System". The paper was published by a scholar using the pseudonym "Satoshi Nakamoto" in 2008. Blockchain is a novel distributed ledger technology that combines P2P (Peer-to-Peer) networks, smart contracts, consensus mechanisms, cryptography, and other techniques [22–24]. The first block in a blockchain is known as the genesis block. Starting from the genesis block, transaction blocks with timestamps and hash values are linked together in chronological order. Each block records the hash value of its parent block, and contains all transaction information within the current time period. Blocks that have been verified by other nodes in the network are recorded in the blockchain and cannot be modified further. The structure of a blockchain is illustrated in Figure 2.

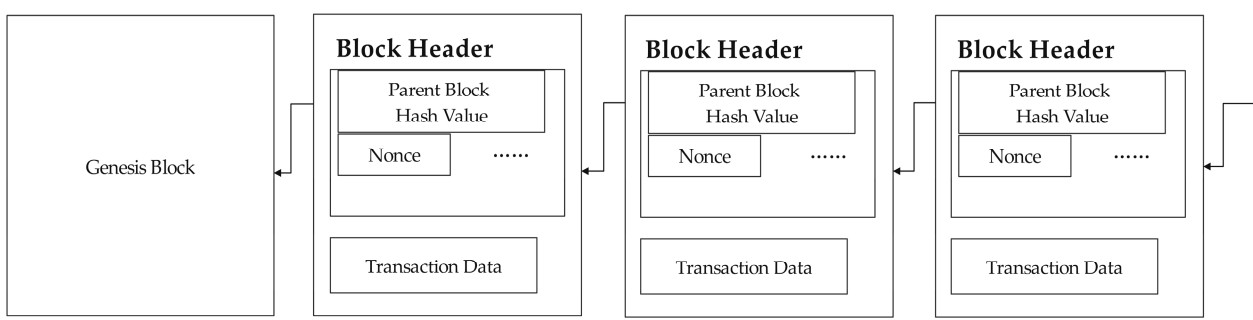

**Figure 2.** Blockchain structure.

### 2.3. PBFT Consensus Algorithm

The PBFT (Practical Byzantine Fault Tolerance) consensus algorithm is a Byzantine fault-tolerant algorithm based on state machine replication, designed for achieving consensus in distributed systems. This algorithm primarily consists of two protocols: the consistency protocol and the view-change protocol. Nodes are categorized into three roles: client, leader, and consensus node. It allows for a maximum of $F = (N - 1)/3$ Byzantine nodes, where N is the total number of nodes in the network and F represents the number

of Byzantine nodes in the network. The consistency protocol is divided into five phases: request, pre-prepare, prepare, commit, and reply [25–27], as illustrated in Figure 3.

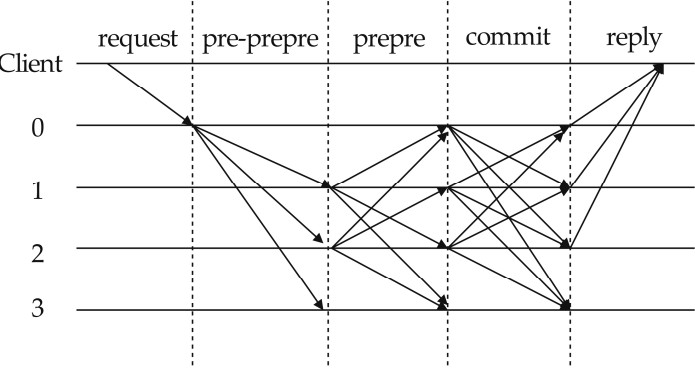

**Figure 3.** PBFT consensus process.

## 3. Blockchain Consensus Model for Vehicular Networks

The blockchain consensus model tailored for vehicular networks, as depicted in Figure 4, consists of three main components: the OBU section, the RSU section, and the blockchain section. Each of these components serves specific functions as outlined below:

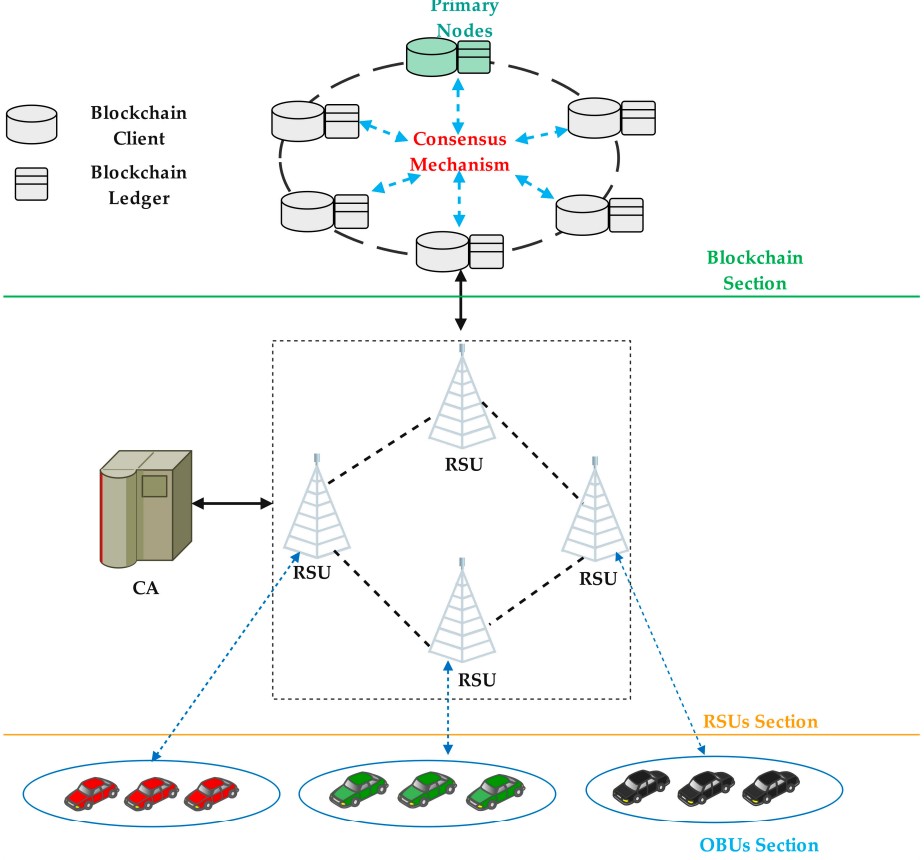

**Figure 4.** Blockchain consensus model tailored for vehicular networks.

OBU Section: Serving as the communication module of vehicles, it interacts with surrounding vehicles by sharing traffic information collected on the vehicle. It follows the principle of proximity and communicates with the nearest RSUs.

RSU Section: This section is responsible for collecting traffic information sent by vehicles and providing services related to vehicle access and authentication. It interacts with

both the blockchain network and vehicle nodes. Additionally, all RSUs collectively maintain a ledger responsible for auditing and verifying the collected transaction information before recording it on the blockchain. Some authorized RSUs can undertake consensus work. Furthermore, RSUs also have the capability to monitor vehicle behavior, assess the credibility of messages sent by vehicles, and update ratings.

Blockchain Section: Based on the proposed blockchain consensus model tailored for vehicular networks, the primary nodes are $\eta_1$, which are selected from the RSU section. These primary nodes possess voting rights, block generation rights, and block verification rights. The remaining RSU nodes are responsible for voting and block verification. In the OBU section, high-quality vehicle nodes are selected as ordinary nodes in the consensus process. These nodes only have voting rights.

Based on this consensus model, the ESBCP consensus process is outlined as follows:

(1) RSUs select high-quality vehicle nodes from the OBU section to participate in consensus based on trust assessment algorithms and an improved algorithm using a unique node list.

(2) Node partition algorithm design is carried out by calculating the similarity between nodes based on communication delay, route hops, and distance.

(3) Consensus nodes in the blockchain section proceed with the corresponding partition's DK-PBFT (Dynamic K-medoids Practical Byzantine Fault Tolerance) consensus process. DK-PBFT includes the Pre-prepare, Prepare1, Prepare2, Commit1, and Commit2 phases.

## 4. ESBCP Consensus Protocol Design and Implementation

To address the challenges posed by the characteristics of large-scale nodes, mobility, and uncertain identities in vehicular network environments to the consensus mechanism in blockchain systems, we propose ESBCP, an efficient and secure blockchain consensus protocol tailored for vehicular networks. The protocol primarily comprises four core algorithms: trust assessment algorithm, node partition algorithm, unique node list improvement algorithm, and DK-PBFT consensus algorithm.

### 4.1. Trust Assessment Algorithm

The trust assessment algorithm evaluates the trustworthiness of vehicles by assessing their trust values. This resolves the issue of unreliable interactions when communicating with different neighboring vehicles [28]. The process is carried out by RSUs.

Definition: The trust value is an assessment of the credibility of the vehicles. It guides the selection of vehicles for communication based on their level of trustworthiness. The trust value of a vehicle node is primarily determined by two components: the global trust value and the V2V trust value. $T$ represents the trust value of a vehicle node, $GT$ represents the global trust value, and $IT$ represents the trust value of the V2V.

Global Trust Value ($GT$): It refers to the trust generation set of the target vehicle in the retrospective time period [0, t]. Assuming r rounds of consensus have occurred within the interval [0, t], the calculation formula for the global trust value can be represented by Formulas (1) and (2).

$$GT(i)_r = \frac{1}{1 + e^{(\zeta \sum\limits_{u=1}^{r} d_u + \xi \sum\limits_{u=1}^{r} \omega_u - \psi \sum\limits_{u=1}^{r} \chi_u - \varsigma \sum\limits_{u=1}^{r} c_u)}} \tag{1}$$

$$GT(i)_{cur}^{acc} = (1 - \updownarrow)GT(i)_r + \updownarrow GT(i)_{cur-1}^{acc} \tag{2}$$

where $d_u$ represents the online status of vehicular network nodes in the $u$-th consensus round, where 1 signifies offline and 0 signifies online. $\zeta$ is the penalty coefficient for offline vehicular network nodes. $\omega_u$ represents the number of invalid transactions submitted by vehicular network nodes in the $u$-th consensus round. $\xi$ is the penalty coefficient of malicious transactions. $\chi_u$ represents the number of legitimate transactions submitted by

vehicular network nodes in the $u$-th consensus round. $\psi$ is the reward coefficient of real transactions. $c_u$ represents the participation outcome of the vehicular network nodes in the $u$-th consensus round, where 1 denotes success and 0 denotes failure. $\varsigma$ is the weight coefficient for rewards granted after consensus success. $GT(i)_r$ represents the global trust value of the $i$-th node in the $r$-th consensus round. $GT(i)_{cur}^{acc}$ represents the current global trust value of vehicle node $i$. $GT(i)_{cur-1}^{acc}$ represents the historical global trust value of node $i$. $\updownarrow$ represents the weight coefficient of historical trust values.

Interactive Trust Value (IT) in V2V: It refers to the comprehensive rating provided by all vehicles within the communication range of the RSUs, based on their interactions with the target vehicle during the time interval [0, t].

The calculation formula for IT can be represented by Formula (3).

$$IT(m,i,t) = \frac{\sum_{\lambda_\mu \in \gamma_g(m,i,t)} y(\lambda_\mu) \cdot f_g}{\sum_{\lambda_\mu \in \gamma_g(m,i,t)} y(\lambda_\mu)} \tag{3}$$

where $\lambda_\mu$ represents the $\mu$-th interaction event, $\gamma_g(m,i,t)$ represents all interaction events collected by RSU $m$ about target vehicle $i$ during time $t$, and $y(\lambda_\mu)$ represents the deviation coefficient of the evaluation of event $\lambda_\mu$ by the vehicles.

The trust value of the $i$-th vehicle node is denoted as $T(i)$. The calculation formula for $T(i)$ can be expressed using Formula (4).

$$T(i) = \frac{\delta_G \cdot GT(i)_{cur}^{acc} + \delta_I \cdot IT(m,i,t)}{\sum_{X \in (G,I)} \delta_X} \tag{4}$$

where $G$ and $I$ respectively represent global trust and V2V trust. $X$ represents one of $G$ and $I$. $\forall X \in (G,I)$, $\delta_X \in [0, 1]$; $\delta_G$ represents the reliability coefficient of the global trust value, while $\delta_I$ represents the reliability coefficient of the V2V trust value.

The pseudocode of the trust value calculation algorithm is shown in Algorithm 1.

---

**Algorithm 1:** Trust Value Calculation

---

**Input**: $(\lambda_p, L[])$, $GT(i)_{cur-1}^{acc}$, $t$, $t_1$
**Output**: $T(i)$
1:      **set** $T(i)_0^{acc} \leftarrow 0$; $Num \leftarrow count(L)$; $r \leftarrow \frac{t}{t_1}$;
2:          **for** $i = 0; i < Num; i{+}{+}$ **do**
3:              **for** $u = 1; u < r + 1; u{+}{+}$
4:                  $compute\ GT(i)_r$;
5:              **if** $GT(i)_{cur-1}^{acc} = 0$ **then**
6:                      $GT(i)_{cur}^{acc} \leftarrow GT(i)_r$;
7:                  **else**
8:                      $compute\ GT(i)_{cur}^{acc}$;
9:                  **end if**
10:             **end for**
11:              $Receive(\gamma_j(m,i,t))$;
12:             $compute\ IT(m,i,t)$;
13:             $compute\ T(i)$;
14:         **end for**

---

According to Algorithm 1, vehicle $i$ can register with the Certificate Authority (CA) to obtain the initial global trust value, denoted as $T(i)_0^{acc}$. During the process of information sharing, the real-time behavioral attributes of a vehicle node are defined as set $RB = \{d, \omega, \chi, c\}$. This mechanism utilizes $RB$ to dynamically generate a trust value, denoted as $T(i)_{cur}^{acc}$ for vehicle node $i$. This facilitates the prompt acquisition of the trust value of senders, denoted as $T(i)_{cur}^{acc}$. Additionally, the V2V trust value, denoted as $IT(m,i,t)$, is computed based on the interaction event $\gamma_j(m,i,t)$ collected by RSUs. Finally, the vehicle trust value, denoted as $T(i)$, is derived from both the global trust value and V2V trust value.

As vehicles continue to send messages, $T(i)$ will also undergo continuous changes. This system efficiently achieves trust evaluation, effectively reducing the presence of malicious nodes. In this mechanism, vehicle nodes are not involved in the calculation process, they only engage in message communication. RSUs verify the messages sent by vehicle nodes, and the server completes the corresponding trust value calculation. The trust values of vehicle nodes are transmitted, stored, and maintained within the blockchain network.

*4.2. Node Partition Algorithm*

In the IoV, there are large numbers of vehicles and devices involved. By partitioning the nodes, consensus tasks can be processed in parallel. The consensus process in different partitions can proceed simultaneously, thus accelerating the overall consensus process and improving its efficiency. The optimization of node partition adapts the consensus algorithm to the specific environment and requirements of the IoV, ensuring an efficient and stable consensus process within the complex IoV network.

K-medoids clustering, an optimization of the k-means clustering algorithm, is a partition-based unsupervised clustering algorithm [29–31]. The k-medoids clustering algorithm attempts to iterate through all sample points within the current cluster, calculating the sum of distances to other sample points, and then selects the optimal point as the new center. To better align the k-medoids clustering algorithm with the PBFT consensus algorithm, efforts are made to assign nodes with lower communication latency and closer distances to the same cluster. Therefore, the similarity between nodes is calculated based on communication latency, routing hop count, and distance.

The communication latency between nodes can be represented by Formula (5).

$$T = \begin{bmatrix} t_{11} & t_{12} & \cdots & t_{1n} \\ t_{21} & t_{22} & \cdots & t_{2n} \\ \vdots & \vdots & \ddots & \vdots \\ t_{n1} & t_{n2} & \cdots & t_{nn} \end{bmatrix} \tag{5}$$

where $T$ represents the matrix of communication latency between nodes, $t_{i,j}$ represents the communication latency between nodes $i$ and $j$ (when $i$ equals $j$, $t_{ij}$ denotes the latency between the node and itself, which is 0), and $n$ represents the total number of vehicle nodes.

Similarly, the routing hop count between nodes can be expressed using Formula (6).

$$R = \begin{bmatrix} r_{11} & r_{12} & \cdots & r_{1n} \\ r_{21} & r_{22} & \cdots & r_{2n} \\ \vdots & \vdots & \ddots & \vdots \\ r_{n1} & r_{n2} & \cdots & r_{nn} \end{bmatrix} \tag{6}$$

where $R$ represents the matrix of routing hop count between nodes, and $r_{i,j}$ represents the routing hop count between nodes $i$ and $j$ (when $i$ equals $j$, $r_{i,j}$ denotes the hop count between the node and itself, which is 0).

Similarly, the distance between nodes can be expressed using Formula (7).

$$D = \begin{bmatrix} d_{11} & d_{12} & \cdots & d_{1n} \\ d_{21} & d_{22} & \cdots & d_{2n} \\ \vdots & \vdots & \ddots & \vdots \\ d_{n1} & d_{n2} & \cdots & d_{nn} \end{bmatrix} \tag{7}$$

where $D$ represents the matrix of distances between nodes, and $d_{i,j}$ represents the distance between nodes $i$ and $j$ (when $i$ equals $j$, $d_{ij}$ denotes the distance between the node and itself, which is 0).

The specific steps for clustering $n$ vehicle nodes into $k$ groups are as follows:

Step 1: Set the current iteration count NI = 0. Select the initial $k$ nodes with the highest trust values from $n$ vehicle nodes.

Step 2: Utilize Formula (8) to calculate the similarity between each non-central point and all central points. Simultaneously, apply the proximity principle to cluster the nodes, thereby forming a collection of clusters, denoted as $\{C_1, C2, \ldots, C_k\}$.

$$Dist(\overline{X_i}, \overline{Y_j}) = \sqrt{t_{ij} r_{ij} d_{ij}} \tag{8}$$

Step 3: Successively, within each cluster, non-central nodes take turns replacing the central node, and the total cost resulting from this substitution is calculated using Formulas (9) and (10). When Cos$t$ is less than 0, the node is used to replace the central point; otherwise, it remains unchanged.

$$E = \sum_{i=1}^{k} \sum_{p \in C_i} |p - q_i|^2 \tag{9}$$

$$\text{Cos}t = E_{cur} - E_{pre} \tag{10}$$

where $p$ represents the non-central points within the cluster, $q_i$ denotes the central point of cluster $C_i$, and $E$ signifies the sum of squares of deviations of all nodes within the cluster.

Step 4: After updating the initially generated cluster set and obtaining the optimal cluster set $\{C_1, C2, \ldots, C_k\}$ composed of the best $k$ central points, we partition the remaining vehicle nodes based on the nearest principle. If there are no updates to the cluster set, skip Step 4. If there are too few nodes in a partition, expand the partition range. Conversely, if there are too many nodes in a partition, reduce the partition range. Ensure that the number of nodes in each partition remains within a specified range.

*4.3. Improved UNL Algorithm*

In applications combining blockchain and the IoV, trustworthiness plays a crucial role. The concept of UNL from the Ripple Protocol Consensus Algorithm (RPCA) can be introduced to enhance the trustworthiness of nodes. However, the original UNL in the RPCA consensus algorithm has some limitations, such as its static nature, meaning nodes cannot be changed once added, and its lack of adaptability, as it cannot dynamically respond to the dynamic changes of vehicle nodes in the IoV [32–36]. To address these issues, we introduce the concept of a Dynamic Unique Node List (DUNL), allowing nodes in the IoV to dynamically select or exclude nodes from the UNL based on their trust values, thus filtering out more reliable nodes for inclusion in the UNL. Building on the previously discussed node partition and trust assessment mechanism, we enhanced the UNL in the RPCA consensus algorithm. Initially, vehicle nodes in their respective partitions exclude any abnormal nodes. Subsequently, they calculate the reliability of nodes in the list of communicable nodes. Based on this reliability, nodes are selected in descending order of trustworthiness to join the UNL, thereby improving the UNL in the RPCA consensus algorithm. The specific formula for calculating reliability is shown in Formula (11).

$$R_{i,j} = (l_1 \times T_{i,j} - l_2 \times v_{i,j} - l_3 \times \tau_{i,j} - l_4 \times \varphi_{i,j}) \times t_{i,j} \tag{11}$$

where $R_{i,j}$ represents the reliability of the $j$-th node in the list of communicable nodes of the $i$-th vehicle node. $T_{i,j}$, $v_{i,j}$, $\tau_{i,j}$, $\varphi_{i,j}$ respectively denote the trust value, offline time, number of invalid transactions provided, and delay time of the $j$-th node in the list of communicable nodes of the $i$-th vehicle node. $l_1$ represents the weight coefficient for trust value, $l_2$ represents the weight coefficient for offline time, $l_3$ represents the weight coefficient for the number of invalid transactions provided, and $l_4$ represents the weight coefficient for delay time. $t_{i,j}$ represents the remaining online time of the $j$-th node in the list of communicable nodes of the $i$-th vehicle node. In case of equal reliabilities, one is randomly selected to be added to the UNL. Therefore, when $i$ vehicle detects event $\lambda_\mu$, it

sends the event message to the vehicle nodes in the UNL and reports it to the nearest RSUs, reducing event verification time and effectively reducing the likelihood of congestion.

### 4.4. DK-PBFT Consensus Algorithm

The vehicle nodes in the blockchain network, after undergoing authorization and verification by the CA, join the blockchain network. Simultaneously, they will have trust values and a DUNL. Both the trust values and DUNL will undergo dynamic updates.

In this paper, the nodes participating in the IoV are categorized based on the roles they assume and the functions they perform within the system, as outlined in Table 1.

**Table 1.** Node descriptions.

| Node Type | Node Description | Node Rights |
|---|---|---|
| Primary Node | Trusted, selected from RSUs | Voting Rights, Block Generation Rights, Block Verification Rights |
| General RSU Node | Trusted | Voting Rights, Block Verification Rights |
| Normal Node | Vehicle nodes ensuring correct and timely message communication | Voting Rights |
| Abnormal Node | Experiencing malfunction or engaged in malicious behavior | No rights granted |

Due to the inherent high trustworthiness of RSUs, a select number of RSUs are chosen as the primary nodes of the consensus group, while the remaining nodes function as regular nodes. The DK-PBFT consensus algorithm, in order to enhance the unpredictability of primary node selection, employs a Verifiable Random Function (VRF) to select multiple primary nodes from the RSUs. Within a partition, the DK-PBFT encompasses a total of N nodes, with $\eta_1$ being primary nodes. The client, serving as the initiator of the request, broadcasts the request message to the blockchain network in the form of a transaction request during the request phase. The primary nodes primarily handle the reception of request messages and sort transactions $m_s$ based on timestamps.

Upon receiving a request from the client, the primary nodes engage in the consensus process with the other consensus nodes, as illustrated in Figure 5. Client represents the client; nodes 0 and 5 are primary nodes; nodes 1, 4, and 8 are normal nodes; nodes 2 and 7 are general RSUs nodes; and nodes 3, 6, and 9 are abnormal nodes.

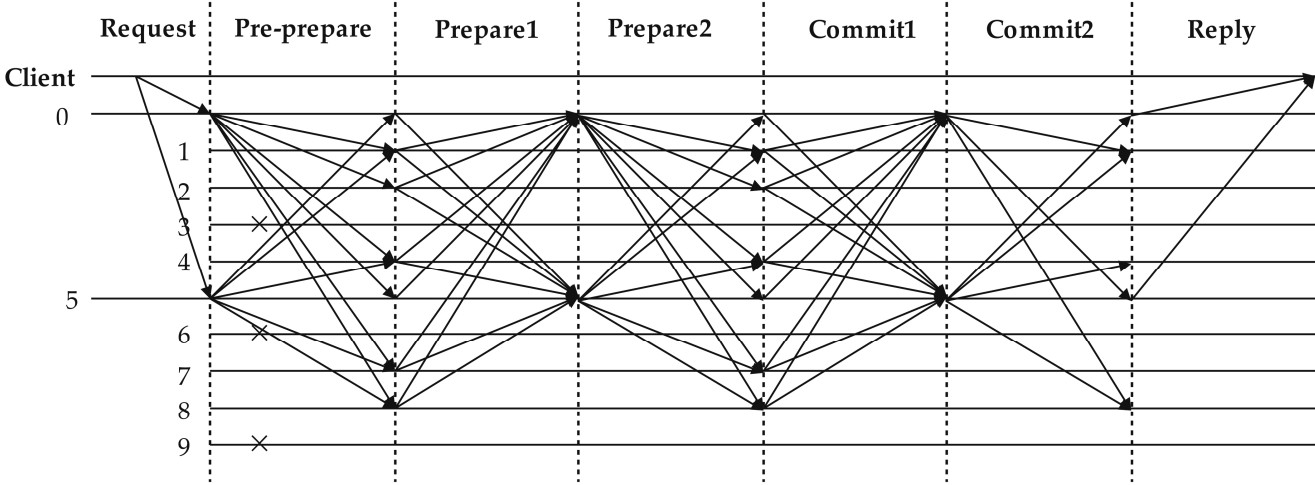

**Figure 5.** Process of the DK-PBFT consensus algorithm.

After implementing the node partition algorithm, each partition can independently engage in the DK-PBFT algorithm consensus. The pseudocode for the DK-PBFT consensus algorithm is presented in Algorithm 2, and the main process of the optimized DK-PBFT is illustrated in Figure 5. The DK-PBFT consensus mechanism encompasses five phases to complete a round of consensus:

(1) Pre-prepare Phase: After verifying the signature of the received request information $< Request, o, t, sign(o), s >$, each primary node broadcasts a pre-prepare message $<< Pre - prepare, sign(m_s), s >, m_s >$ to other primary nodes. In this message, $Pre - prepare$ identifies the pre-prepare message for consensus, $sign(m_s)$ represents the signature of the primary node on pre-prepare message $m_s$, $s$ is the sequence number assigned by the primary nodes to message $m_s$, and message comprises the original transaction request set with the signature of the client. Primary nodes also broadcast pre-prepare messages to the participating consensus nodes in their respective vicinity.

(2) Prepare1 Phase: Upon receiving the Pre-prepare message from the primary nodes, the consensus nodes first verify the signature $sign(m_s)$ and sequence number $s$ of the message. After successful verification, it takes the union of transactions $m_s$ from different primary nodes and sorts them based on timestamps. The resulting transaction set is denoted as $M$. The consensus node then sends a message $<< Prepare1, H(M), sign(H(M)) >, M >$ to the primary nodes, where $H(M)$ is the hash value of transaction set $M$.

(3) Prepare2 Phase: When the primary nodes receive the Prepare1 message from more than $\frac{2N}{3}$ nodes, it compares the hash values from each message. If the hash values from more than $\frac{N}{3}$ nodes are the same, the primary nodes broadcast a message $<< Prepare2, Prepare1set, sign(H(M) \parallel Prepare1set >, Prepare1set >$ to all nodes, where $Prepare1set$ represents the collection of $Prepare1$ messages received by the primary nodes.

(4) Commit1 Phase: After receiving the Prepare2 message, normal nodes vote on the message and then send the voting information $Vote(Prepare2)$ back to the primary nodes.

(5) Commit2 Phase: When the master node receives the Commit1 message from over $\frac{2N}{3}$ nodes, it performs a weighted calculation to decide whether to add the information $Vote(Prepare2)$ shared by these vehicle nodes to the blockchain. After successful verification, the primary nodes package these transactions into a block. The primary nodes broadcast this block to all RSU nodes. When the block is validated by all consensus nodes, it signifies the completion of consensus and successful blockchain integration.

RSUs update the trust values and DUNL based on the consensus results.

---

**Algorithm 2:** DK-PBFT

---

**Input:** *Request*
**Output:** *result*
1:  $NodePartition()$;
2:  **while** $< Request, o, t, sign(o), s >= True$ **do**
3:      broadcast $<< Pre - prepare, sign(m_s), s >, m_s >$;
4:      **if** $< Pre - prepare >= True$
5:          $order(m_s)$;
6:          $M = m_1 \cup m_2 \cup \cdots \cup m_s$;
7:          send $<< Prepare1, H(M), sign(H(M)) >, M >$;
8:      **end if**
9:      **if** $Total(Prepare1) > \frac{2N}{3}$
10:        **if** $Total(samehash) > \frac{N}{3}$
11:            send $<< Prepare2, Prepare1set, sign(H(M) \parallel Prepare1set >, Prepare1set >$;
12:        **end if**

---

```
13:        end if
14:        if Receive(Prepare2) = True
15:          Vote(Prepare2);
16:        end if
17:         if Total(Commit1) > 2N/3
18:           Update(T(i));
19:         end if
20:         if < Commit2 >= True then
21:         do < Request, o, t, sign(o), s >;
22:         reply to the client;
23:         end if
24:         VerifyBlock();
25:         chain(block);
26:         TrustValueCalculation();
27:         DUNL();
28:          return result;
29:    end while
```

## 5. Theoretical Analysis

In this section, we will analyze ESBCP from two aspects: security and communication overhead.

### 5.1. Security Analysis

In the scenario of combining blockchain with the IoV, security is of importance. If the blockchain were to be subjected to an attack, it could lead to the exposure of user information, vehicle location data, and other private data, potentially even causing serious consequences such as traffic accidents. In this context, we primarily consider two types of security risks. Firstly, there is the external security risk, where non-members of the IoV may attempt to impersonate existing nodes within the IoV to deceive the system and become part of the network. Secondly, there is the internal security risk, where properly registered and validly signed IoV nodes may become potential security vulnerabilities due to malicious attacks.

To address these two security risks, we employed a permissioned chain mechanism in the constructed blockchain. Every vehicle intending to participate in the IoV must be approved by the CA before being able to join the IoV and utilize shared data. Upon joining the IoV, each vehicle entity possesses its own public–private key pair, and any transaction or message requires the signature of its private key. In this way, when a vehicle receives any message or transaction, the system first verifies if the signature is correct and simultaneously checks if the corresponding public key address is in the permission list. If the signature is incorrect or the sending vehicle is not in the permission list, the message sent will be rejected, effectively preventing these two security risks.

ESBCP further safeguards the security and reliability of the IoV by restricting participation in the consensus to authorized and legitimate vehicles. This approach effectively hinders potential attacks, ensuring the integrity and accuracy of the IoV consensus. In the dynamically changing environment of the IoV, by partitioning nodes and selecting high-quality nodes to participate in the consensus, we can ensure that the consensus algorithm operates efficiently and stably in the complex IoV network.

### 5.2. Communication Overhead Analysis

Traditional PBFT consensus algorithms consume significant communication resources. To comprehensively evaluate the performance of ESBCP, let us assume there are a total of N nodes in the IoV, with $\eta$ being the number of RSU nodes, and $\eta_1$ being the number of primary nodes. As shown in Figure 5, the DK-PBFT consensus algorithm is divided into five phases. In the pre-prepare phase, each primary node broadcasts the pre-prepare message, resulting in a maximum communication volume of $\eta_1(N - 1)$ in this phase. In

the two prepare phases, vehicle nodes communicate with primary nodes, resulting in a total communication volume of up to $2\eta_1(N-1)$. In the first commit phase, vehicle nodes communicate with primary nodes, resulting in a total communication volume of up to $2\eta_1(N-1)$. In the second commit phase, primary nodes communicate with other RSU nodes, with a maximum communication volume of $\eta_1(\eta-1)$.

In this paper, we list the communication overhead required for one round of consensus for the ESBCP, PBFT, SG-PBFT, and CDBFT consensus algorithms according to the consensus stages, as shown in Table 2. It can be observed that the communication complexity of the PBFT, SG-PBFT, and CDBFT consensus algorithms are all $O(N^2)$, while the consensus mechanism proposed in this paper has a communication complexity of $O(N)$. This indicates that the communication overhead of the DK-PBFT consensus algorithm proposed in this paper is much lower than that of the traditional PBFT, SG-PBFT, and CDBFT. Although the DK-PBFT adds two phases and requires RSU nodes to verify blocks in later stages, which consumes additional communication resources, it enhances the security of the system.

**Table 2.** Byzantine consensus algorithm communication overhead comparison.

| Consensus Phase | PBFT | CDBFT | SG-PBFT | ESBCP |
|---|---|---|---|---|
| Request | 1 | 1 | 1 | $\eta_1$ |
| Pre-prepare | $N-1$ | $N-1$ | $N/2-1$ | $\eta_1(N-1)$ |
| Prepare | $(N-1)^2$ | $(N-1)^2$ | $(N/2-1)(N/2-1)$ | $2\eta_1(N-1)$ |
| Commit | $N(N-1)$ | $N(N-1)$ | $N/2-1$ | $\eta_1(N-1)+\eta_1(N-1)$ |
| Reply | $N$ | $N$ | $N$ | $\eta_1$ |
| Total | $2N^2-N+1$ | $2N^2-N+1$ | $N^2/4+N$ | $4\eta_1 N-3\eta_1+\eta_1\eta$ |

## 6. Experimental Analysis

To comprehensively evaluate the performance of ESBCP, we conducted simulated experiments and compared the performance with existing CDBFT algorithms. The simulation experiments utilized Go to simulate the consensus process of the improved consensus algorithm. The hardware environment consisted of an AMD 2.10 GHz Ryzen 5 5500U processor and 16 GB of memory, running on a 64-bit Windows 11 operating system. The speed of vehicular network nodes, denoted as 'mv', was 12 m per second (12 m/s). These nodes maintained a constant speed throughout the simulation. Dynamic routing was employed in the experiment. This process was simulated without specific content, focusing solely on aspects crucial for the blockchain component. The following assumptions were made about the network environment: (1) sufficient computational power for RSUs; (2) malicious nodes in the network are less than 1/3; (3) the total number of nodes in the network is N, with the actual nodes participating in consensus denoted as N1. Let N1/N = a, where a represents the ratio of actual nodes participating in consensus to the total number of nodes in the network. Considering the differences in transaction quantity per block, input parameters for concurrent transactions by vehicle nodes, and road conditions in each experiment, we conducted each experiment 10 times, taking the average as the final result to reduce errors.

### 6.1. Consensus Latency

Consensus latency refers to the total duration from the request phase to the reply phase in communication. A consensus algorithm with shorter latency leads to faster transaction confirmation speeds, thus leading to higher efficiency in vehicular networks. In this paper, the consensus latency of the two algorithms was compared under the same latency environment, with node quantities of 100, 200, 300, 400, 500, and 600 tested to observe the impact of node quantity on consensus latency. As shown in Figure 6, with an increase in node quantity, the CDBFT consensus algorithm exhibits an upward trend in consensus latency. In contrast, ESBCP, through a trust evaluation mechanism, comprehensively considers the performance of vehicle nodes and reasonably allocates rights. Additionally, each vehicle node maintains a unique node list, thereby excluding nodes with poor performance or

malicious behavior. Ultimately, this stabilizes and maintains the consensus latency at a relatively low and stable level. The CDBFT consensus algorithm only selects nodes to construct blocks based on reputation values without a comprehensive evaluation of node performance. Therefore, the consensus latency of ESBCP is superior to that of the CDBFT consensus algorithm.

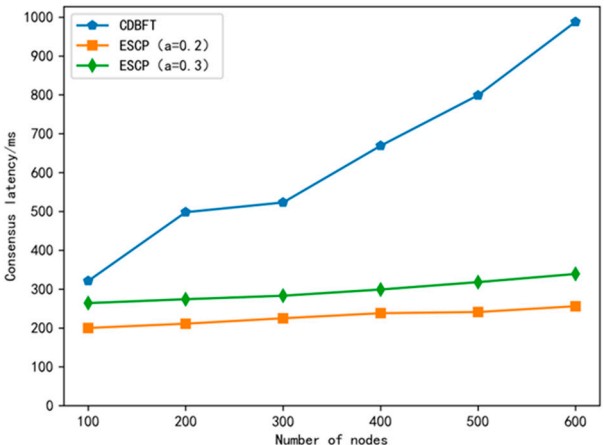

**Figure 6.** The impact of node quantity on consensus latency.

### 6.2. Throughput

Throughput, measured in Transactions Per Second (TPS), refers to the quantity of transactions processed by a network within a given unit of time. It quantifies the capability of the consensus algorithm to handle transactional data. The magnitude of TPS reflects the efficiency of this process. The definition of throughput is illustrated in Formula (12).

$$TPS = \frac{SUM}{pe} \tag{12}$$

where *pe* represents the time interval, while *SUM* signifies the number of transactions processed within time *pe*.

In this paper, a comparison of the throughput of the two algorithms was conducted under the same latency environment, with node quantities set at 100, 200, 300, 400, 500, and 600. As shown in Figure 7, the throughput of the CDBFT consensus algorithm exhibits a declining trend as the number of vehicle nodes increases. This is attributed to the significant increase in network bandwidth pressure due to extensive communication overhead between nodes, leading to a notable increase in consensus latency and subsequently a decrease in throughput. ESBCP maintains a relatively stable throughput with a slight decline as the number of nodes increases. This is attributed to the parameter 'a' being set, resulting in a gradual growth of participating vehicle nodes in consensus, which remains at a constant quantity. This significantly alleviates the bandwidth pressure from inter-node communications. Additionally, ESBCP introduces a trusted node list and proposes node partition strategies based on communication latency, routing hops, and distance between nodes to mitigate the impact of offline nodes or malicious behavior on block consensus efficiency. While the CDBFT consensus algorithm completes block consensus through representative nodes, it does not account for the impact of representative vehicle nodes going offline. Therefore, the throughput of the ESBCP consensus algorithm surpasses that of CDBFT.

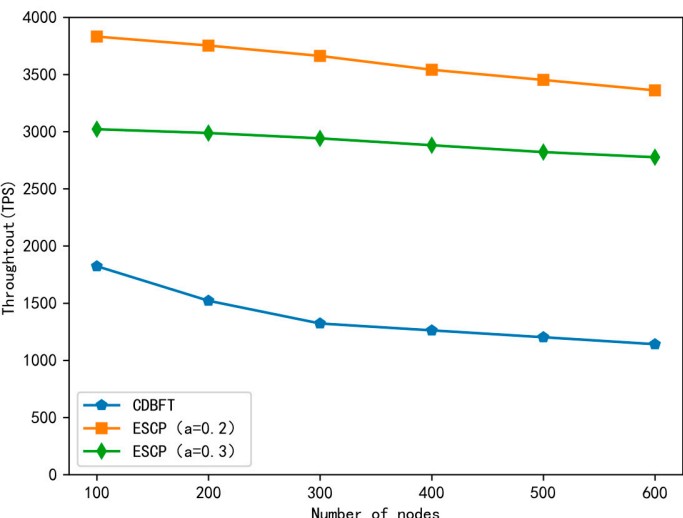

**Figure 7.** The impact of node quantity on throughput.

### 6.3. Malicious Node Precision

In this paper, Malicious Node Precision (MNP) refers to the proportion of vehicle nodes identified as malicious among those with low reliability and trust values, utilizing the credit assessment mechanism and dynamic unique node list. It serves as a metric to gauge the accuracy of the vehicular network system in identifying malicious nodes. The calculation formula for MNP in vehicular networks is shown in Formula (13).

$$MNP = TP/(TP + FP) \tag{13}$$

where TP (True Positive) represents the number of correctly predicted malicious nodes, and FP (False Positive) represents the number of normal nodes incorrectly predicted as malicious.

By selecting node quantities of 100, 200, 300, 400, 500, and 600, with malicious node ratios set at 5%, 10%, 15%, and 20%, the analysis of the impact of malicious node quantity on the precision of ESBCP is presented in Figure 8. It is observed that ESBCP consistently maintains the MNP over 90%. Building upon the PBFT consensus algorithm, the integration of trust assessment mechanisms and UNL effectively detects malicious nodes, thereby enhancing the quality of participating consensus nodes. The trust values between normal and malicious nodes are distinctly differentiated. Consequently, as both the node quantity and malicious node ratio increase, the precision of ESBCP exhibits a slight upward trend.

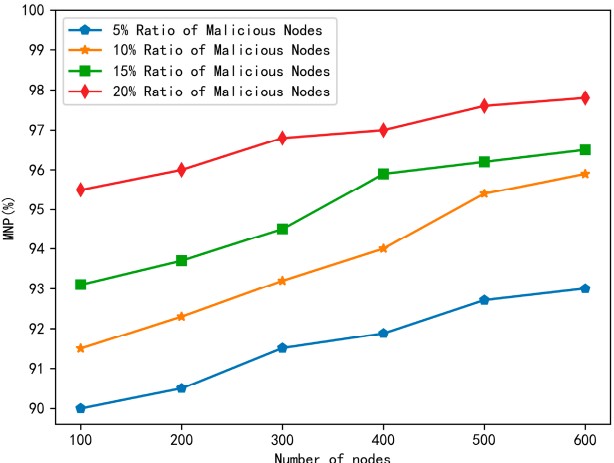

**Figure 8.** The impact of malicious node ratio on precision.

*6.4. Malicious Node Recall*

In this paper, Malicious Node Recall (MNR) refers to the proportion of malicious nodes detected among vehicle nodes with low reliability and trust values, utilizing the credit assessment mechanism and DUNL. It measures the performance of system in discovering all malicious nodes. MNR can be calculated using Formula (14).

$$MNR = TP/(TP + FN) \tag{14}$$

where FN (False Negative) represents the number of samples that are actually malicious nodes but incorrectly predicted as normal nodes.

By selecting node quantities of 100, 200, 300, 400, 500, and 600, with malicious node ratios set at 5%, 10%, 15%, and 20%, the analysis of the impact of malicious node ratio on the MNR of ESBCP is presented in Figure 9. It is observed that as the number of nodes in the network increases, the MNR of ESBCP exhibits an upward trend. However, with an increase in the proportion of malicious nodes, the MNR of ESBCP shows a slight decline. As the number of nodes participating in the network increases, the trust assessment mechanism of ESBCP and DUNL effectively differentiate between normal and malicious nodes, resulting in an increase in MNR. Additionally, as the proportion of malicious nodes gradually increases, ESBCP completes block validation consensus through RSUs, preventing malicious nodes participating in consensus from having the opportunity to launch attacks. This results in their performance during consensus being similar to that of normal nodes, categorizing them as general nodes. As a result, the MNR of ESBCP shows a slight decrease.

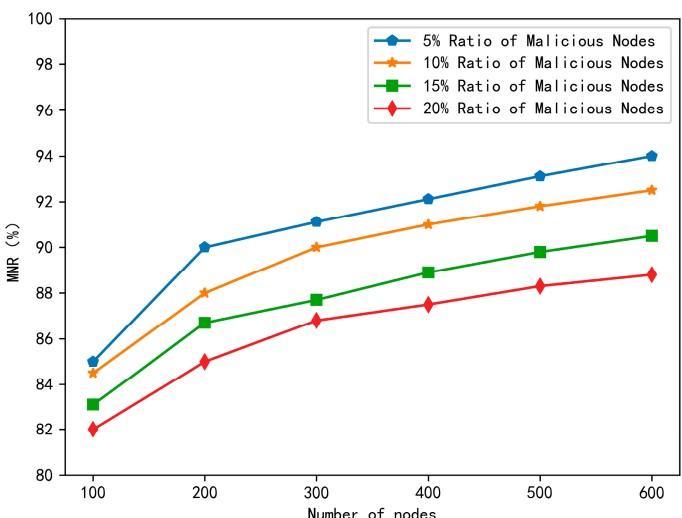

**Figure 9.** The impact of malicious node proportion on MNR.

## 7. Conclusions

In this paper, we propose ESBCP, an efficient and secure consensus protocol for blockchain tailored to the IoV. ESBCP features lightweight characteristics and is well-suited for resource-constrained IoV environments, especially in scenarios where vehicle nodes have limited computational and communication capabilities, demonstrating high operational efficiency. By integrating IoV node trust assessment algorithms, node partition algorithms, dynamic unique node list algorithms, and the DK-PBFT consensus algorithm, this protocol effectively reduces consensus latency and enhances system throughput. The theoretical analysis indicated that ESBCP can effectively prevent external and internal security risks, improving the overall security of the IoV system while ensuring good scalability. Experimental validation demonstrated that ESBCP significantly increases throughput and reduces consensus latency.

**Author Contributions:** Conceptualization, X.S. and M.L.; Data curation, X.S. and J.L.; Formal analysis, Z.Y. and J.L.; Funding acquisition, X.S.; Investigation, M.L. and W.Z.; Methodology, Z.Y. and W.Z.; Project administration, X.S.; Resources, X.S.; Software, M.L.; Supervision, J.L.; Validation, M.L. and Z.Y.; Writing—original draft, M.L.; Writing—review & editing, Z.Y. and Q.Z. All authors have read and agreed to the published version of the manuscript.

**Funding:** This research was funded by the Major Public Welfare Project of Henan Province, grant number 201300210300; Key Science and Technology Project of Henan Province, grant number 222102210168; Open Fund of Henan Key Laboratory of Network Cryptography Technology, grant number LNCT2021-A14, LNCT2022-A12; Songshan Laboratory Pre-research Project, grant number YYJC032022021; and Postgraduate Research and Innovation Project of Zhongyuan University of Technology, grant number YKY2023ZK52.

**Data Availability Statement:** The source code in this study is available upon request from the corresponding author.

**Conflicts of Interest:** The authors declare no conflict of interest.

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
