# Peer review of "An Efficient and Secure Blockchain Consensus Protocol for Internet of Vehicles"

_electronics, doi:10.3390/electronics12204285_

Round 1
Reviewer 1 Report
The paper is well structured and written. Explanations are mostly clear. There are few clarifications and minor issues that should be done/corrected:
- In related work - the first paragraph, second type is mentioned twice, and the latter mentioning should refer to the third type (in sentence: The second type is enhancing the consensus structure).
- In several places ESCP is used instead of ESBCP. Please correct this.
- Text in figure 1 should be more visible. Currently the font of that text is too small.
- In 4.2 clustering process - explanation of grouping n nodes into k groups through steps needs to be improved and clarified. Following is not completely clear: step 2 creation of k clusters - based on (8) it seems that clusters can be unevenly populated - if that is indeed the case, comment on how creation of very uneven populated clusters can affect the overall performance should be given, otherwise if clusters are evenly populated, please add explanation how is this achieved; in step 3 - it is not clear is this step done per cluster in sense that nodes cannot move from one cluster to another (given step 2 clusterization result) when some initial central node is replaced, it seems that step 3 is done for each cluster separately but it is not completely clear from the given text; step 4 - is this step 2 again but with updated set of central points done in step 3 and can this step be skipped if step 3 has not perform any updates.
- In 5.1 you use term permission list but is not defined prior. Please define it. Also, does the given analysis cover all inside attacks. For example, some node is infected with malicious code, and it operates seemingly ok but feeds wrong information during intercommunication with other nodes, for example, some traffic information is poisoned. Please clarify analysis regarding inside attacks.
- In 5.2 you used both n and N. Please unify notation.
- In 6 experimental setup is not completely explained, and one is not able to reproduce the experiment. For example, are vehicle nodes are moving, in which way and at which speed; what type of routing/networking is used, are actual messages are exchanged in simulation or this process is just simulated without some specific content other than important for blockchain part, how malicious nodes are modeled and detected in simulation... Experimental setup must be clearly and in detail explained so someone can reproduce your experiment setup for evaluation purposes.
Overall writing is good, but there are few errors throughout the paper. Thus, I recommend to proofread it once more. Some examples:
We proposes
assessment of a the
Reviewer 2 Report
#1 The title presentation method needs to be modified. The ESBCP in "ESBCP: An Efficient and Secure Blockchain Consensus Protocol for Internet of Vehicles" is consistent with the Efficient and Secure Blockchain Consensus Protocol and should not be repeated. At the same time, if ESBCP is a proprietary property that is commonly recognized in this field, Nouns are also inappropriate to appear in titles.
#2 The source basis of the mathematical formulas related to the model must be supported by objective academic or practical applications. Therefore, the author must have the source basis of the citation, or the source basis that presents the rationality and feasibility of the functional value relationship that presents the practical application of the relevant specific topic.
#3 Regardless of whether it is a specific model or can present a general model, the author's research topic is an algorithm with practical application value, so it is necessary to provide case data to verify the model calculation, so as not to have an overly subjective understanding.
#4 It is recommended that the author summarizes the similarities and differences between the main contents of past research on related topics and the development algorithm of this model, and at the same time provides the practical application value of this research.
Minor editing of English language required.
Reviewer 3 Report
Summary/Contribution: The main contribution of the work is the proposal of ESBCP, an efficient and secure blockchain consensus protocol for the Internet of Vehicles (IoV) environment. ESBCP addresses the issues of low transaction throughput, high latency, and elevated communication overhead commonly faced by traditional blockchain consensus protocols in IoV. It introduces a trust evaluation mechanism executed by Road Side Units (RSUs) to select high-quality vehicle nodes, a node partition strategy to adapt to the dynamic nature of IoV, and a dynamic unique node list to handle node mobility. These strategies, combined with the DK-PBFT consensus algorithm, improve efficiency, security, and scalability in IoV, as demonstrated through theoretical analysis and experimental verification.
Comments/Suggestions:
1. Clarify the definition of IoV (Internet of Vehicles) in the first sentence for readers who may not be familiar with the acronym.
2. Consider providing a brief overview of the significance of IoV in real-world applications to engage the reader's interest.
3. When mentioning challenges like data security, privacy, and vulnerability of nodes, it would be helpful to provide concrete examples or statistics to illustrate the severity of these issues.
4. Explain why blockchain is particularly well-suited to address these issues in the context of IoV. What specific advantages does blockchain offer?
5. When discussing the impact of incorporating blockchain on system performance, provide more specific information or examples to quantify this impact.
6. Consider grouping related work into subsections for better organization, such as one subsection for "Controlling the Number of Consensus Nodes," another for "Optimizing the Consensus Process," and a third for "Improving Consensus Structures."
7. Provide a brief summary or evaluation of each optimization strategy discussed to help the reader understand the strengths and weaknesses of each approach.
8. It may be helpful to include a table or diagram summarizing the key features of each mentioned consensus protocol for easy reference.
9. Discuss the limitations or challenges faced by the existing optimization approaches, especially in the context of IoV, and how your proposed protocol addresses these challenges.
10. When explaining the phases of your proposed blockchain consensus model, consider providing a high-level diagram or flowchart to help readers visualize the process.
11. Mention the practical applications or scenarios where the ESBCP protocol is expected to have the most significant impact within the IoV environment.
12. Include a statement on the potential implications of your research, such as how it might influence the development of IoV technologies or the adoption of blockchain in this domain.
13. Formal methods can be used to verify the correctness of smart contracts and blockchain codes, which can help to prevent costly errors and security breaches. Therefore, it is important to discuss the use of formal methods in your paper.
14. For this purpose, the authors may include the following interesting references (and others):
a. https://ieeexplore.ieee.org/document/9970534
b. https://ieeexplore.ieee.org/document/8328737
Can be improved
Round 2
Reviewer 1 Report
Listed suggestions and questions have been addressed.
Author Response
Thank you your valuable feedback. the results have been clearly presented.
Reviewer 2 Report
The author, while responding to the reviewers' comments, has mostly provided vague and general responses. In the responses, it is not clearly stated what specific changes have been made to the content and page numbers in the revised manuscript. Therefore, it is difficult for the reviewers to determine what the specific modifications are and how they correspond to the content in the revised manuscript.
The author has developed the model based on specific assumptions and characteristics. However, it's unclear whether the model can be applied in a more general context, and the author has not provided objective references to support this. Furthermore, the explanations and analyses of the case studies lack objective references. Therefore, the practical applicability of the research requires further rigorous exploration by the author.
Minor editing of English language required.
Reviewer 3 Report
The authors considered my comments and suggestions.
Can be improved.
Round 3
Reviewer 2 Report
The author continues to provide hasty and inattentive responses to the following questions:
The author, while responding to the reviewers' comments, has mostly provided vague and general responses. In the responses, it is not clearly stated what specific changes have been made to the content and page numbers in the revised manuscript. Therefore, it is difficult for the reviewers to determine what the specific modifications are and how they correspond to the content in the revised manuscript.
The author has developed the model based on specific assumptions and characteristics. However, it's unclear whether the model can be applied in a more general context, and the author has not provided objective references to support this. Furthermore, the explanations and analyses of the case studies lack objective references. Therefore, the practical applicability of the research requires further rigorous exploration by the author.
Minor editing of English language required.
